# Six-month follow-up of minimally invasive nerve-sparing complete excision of endometriosis: What about dyspareunia?

**Claudio Peixoto Crispi Jr.** [1]* , **Claudio Peixoto Crispi** [1‡], **Bruna Rafaela Santos de Oliveira** [1], **Nilton de Nadai Filho** [1‡], **Fernando Maia Peixoto-Filho** [2‡], **Marlon de Freitas Fonseca** [1,2]

1 Crispi Institute of Minimally Invasive Surgery, Rio de Janeiro, Rio de Janeiro, Brazil, 2 Department of Women's Health—Fernandes Figueira National Institute for Women, Children and Youth Health—Oswaldo Cruz Foundation, Rio de Janeiro, RJ, Brazil

☯ These authors contributed equally to this work.
‡ These authors also contributed equally to this work.
* claudin.jr@gmail.com

## Abstract

### Study objective

To assess individual changes of deep dyspareunia (DDyspareunia) six months after laparoscopic nerve-sparing complete excision of endometriosis, with or without robotic assistance.

### Methods

This preplanned interdisciplinary observational study with a retrospective analysis of intervention enrolled 126 consecutive women who underwent surgery between January 2018 and September 2019 at a private specialized center. Demographics, medical history and surgery details were recorded systematically. DDyspareunia (primary outcome), dysmenorrhea and acyclic pelvic pain were assessed on self-reported 11-point numeric rating scales both preoperatively and at six-month follow-up. Cases with poor prognosis in relation to dyspareunia were described individually in greater detail.

### Results

Preoperative DDyspareunia showed weak correlation with dysmenorrhea (rho = .240; P = .014) and pelvic pain (rho = .260; P = .004). Although DDyspareunia improved significantly (P < .001) by 3 points or more in 75.8% (95%CI: 64.7–86.2) and disappeared totally in 59.7% of cases (95%CI:47.8–71.6), individual analysis identified different patterns of response. The probability of a preoperative moderate/severe DDyspareunia worsening more than 2 points was 4.8% (95%CI: 0.0–10.7) and the probability of a woman with no DDyspareunia developing "de novo" moderate or severe DDyspareunia was 7.7% (95%CI: 1.8–15.8) and 5.8% (95%CI: 0.0–13.0), respectively. In a qualitative analysis, several conditions were hypothesized to impact the post-operative DDyspareunia response; these included adenomyosis, mental health disorders, lack of hormone therapy after surgery,

**Data Availability Statement:** All relevant data are within the paper and its Supporting Information files.

**Funding:** Publication fee was funded by a Brazilian Government Institution (CAPES-PROAP budget – award number 001/2021 VPEIC / Fiocruz - Programa de Pós-graduação em Pesquisa Aplicada à Saúde da Criança e da Mulher do Instituto Nacional de Saúde da Mulher, da Criança e do Adolescente Fernandes Figueira, da Fundação Oswaldo Cruz). Fiocruz provided support in the form of salaries for authors FMPF and MFF, but did not have any additional role in the study design, data collection and analysis, decision to publish, or preparation of the manuscript. We also declare that our study received no funding from any company or institution for the field work and development, all costs for this were due to the authors. The funders had no role in study design, data collection and analysis, decision to publish, or preparation of the manuscript.

**Competing interests:** The authors have declared that no competing interests exist.

colporrhaphy, nodule excision in ENZIAN B compartment (uterosacral ligament/parametrium), the rectovaginal septum or the retrocervical region.

## Conclusion

Endometriosis surgery provides significant improvement in DDyspareunia. However, patients should be alerted about the possibility of unsatisfactory results.

## Introduction

Endometriosis is an endemic condition that is associated with pain and different dysfunctions [1, 2]. Surgical eradication is the treatment of choice to improve health-related quality of life in cases in which medical management has been ineffective for pain relief [3, 4] or in selected cases of endometriosis-related infertility [5]. Extensive resections may be necessary when multiple deep infiltrating lesions occur. Because of the risk of urinary and bowel complications [6, 7] an experienced multidisciplinary team should perform the surgery. The latest guidelines on the practical aspects of surgery for the treatment of deep infiltrating endometriosis were elaborated by a clinical expert consensus panel [8] and efforts to identify and preserve autonomic pelvic nerves whenever possible are recommended [9].

The two pain symptoms most frequently associated with endometriosis are dysmenorrhea and deep dyspareunia, which may occur independently [10]. Although dysmenorrhea and chronic pelvic pain are the clinical manifestations most commonly associated with diminished health-related quality of life [1], deep dyspareunia also is a cardinal symptom of endometriosis [11]. Adolescent and young adult women with endometriosis experienced dyspareunia twice as often than those without endometriosis; painful intercourse has a negative impact on their physical and mental wellbeing [12]. Dyspareunia can be classified as superficial (pain at the vaginal introitus with initial penetration) or deep (occurring within the pelvis with deep penetration). Women with dyspareunia experience orgasm less often, which is correlated with a decrease in reported overall well-being [13]. There are several promising avenues for exploration of the pathophysiology and treatment of deep dyspareunia [14].

Surgery has been shown to decrease pain in some but not all women [15]. When persistent or new pain occurs, the physician usually is ill-prepared to assess and treat the problem, and patients experiencing this deception often feel both distressed and unsatisfied. Comprehensive data are lacking on the proportions of patients who experience little or no pain relief after surgery [16] and it is known there are confounders in the reporting of disease and the pain symptoms [17]. Placing the focus on deep dyspareunia, the main aim of this study was to assess the individual responses to minimally invasive nerve-sparing complete excision of endometriosis at six months of follow-up.

## Material & methods

This preplanned interdisciplinary observational study enrolled 126 consecutive cases of Brazilian women who were admitted to the Crispi Institute for Minimally Invasive Surgery (Rio de Janeiro, RJ, Brazil).

The surgeries described in this study were performed between January 2018 and September 2019 and were the standard-of-care for surgical treatment of endometriosis for infertility and/or pain persisting after medical management. The recommendation for surgery was made at the discretion of the attending gynecologist. Prospective written consent for inclusion in

observational studies was informed and signed by all patients prior to the surgical procedure. These documents are stored at our institute.

The research protocol was approved February 21, 2019 by an institutional review board—the Research Ethics Committee of the Oswaldo Cruz Institute Foundation (CAAE 07885019.8.0000.5269 IFF-FIOCRUZ), which authorized inclusion of patients admitted to the Crispi Institute for Minimally Invasive Surgery since January 2018. Patients who might have declined to take part in the study would have received the same care as the patients who gave their consent to take part in the study.

Demographic, clinical and outcome data were abstracted from the medical records and assessed in a database in December 2020. All patients who have undergone minimally invasive surgery for endometriosis were selected for inclusion in this study at the time of data abstraction from medical records.

The Strengthening the Reporting of Observational Studies in Epidemiology (STROBE) statement [18] was followed to improve the quality of reporting.

### Dyspareunia assessment

Systematically, a thorough assessment of the main painful symptoms associated with endometriosis has been carried out in all patients in the first preoperative visit. As a routine, these symptoms were also evaluated in all patients six months after surgery.

Thus, the severity of deep dyspareunia (primary outcome), dysmenorrhea and acyclic/non-menstrual pelvic pain was assessed on a self-reported 11-point numeric rating scale (NRS) at two time points: prior to surgery during the preoperative evaluation period, and at the six-month follow-up encounter; as is done routinely with all patients at our institute. In order to consider possible variations in pain intensity, study participants were instructed to report their preoperative symptoms as representative of the prior six months. The questions for deep dyspareunia asked: "Have you had pain during sexual intercourse in the last six months? If yes, is this pain at the beginning of penetration or during deep penetration?". Considering the deep vaginal penetration only, the participants could mark the NRS or could check "not applicable", if they have not had sexual intercourse in the prior six months. According to the instrument's scale, deep dyspareunia was hierarchically categorized as none/mild (0–3), moderate/tolerable (4–6), or severe (7–10), as in other studies of dyspareunia in the setting of endometriosis [19, 20]. In this study, an improvement of dyspareunia score by a minimum of 3 points was empirically considered a positive response.

Superficial dyspareunia symptoms were not considered and the term dyspareunia here refers solely to reports of deep dyspareunia, even in subjects who experienced both superficial and deep dyspareunia. The participants were free to report–if they thought it was important–any type of deep vaginal penetration (e.g., with sex toys), and to include both opposite-sex and same-sex sexual interactions.

As recommended [21], we used the two leading endometriosis classifications systems to present a case-by-case description in more detail: the 1997 Revised American Society for Reproductive Medicine classification of endometriosis, (rASRM) [22]—the most commonly used, and the ENZIAN, which is most significantly correlated with the extent of the disease, symptoms, and difficulty and length of surgery [23]. This study included no women in whom the pain symptoms assessed preoperatively were wrongly (presumptively) attributed to endometriosis. The histopathology of tissue collected under visual inspection during laparoscopy confirmed endometriosis in all patients.

As endometriosis is not a uniform condition and dyspareunia symptom intensity varies, women have different expectations regarding how surgery may improve their symptoms.

Thus, the responses to surgery were separately assessed in women with preoperative none/mild dyspareunia (NRS< = 3) and in women with moderate/severe dyspareunia (NRS>3).

## The nerve-sparing surgery

In this series, the preoperative diagnosis of endometriosis involved three steps—medical history, physical examination and magnetic resonance imaging (MRI), and an experienced multidisciplinary team led by the same gynecologist (C.P.C.) performed all surgeries.

The laparoscopic nerve-sparing strategy for complete excision of the endometriotic lesions was based on the nerve-sparing Negrar technique [24]. The standardized surgical approach included the development of avascular spaces, and identification and preservation (as much as possible) of pelvic autonomic fibers, such as those of the inferior mesenteric plexus and superior hypogastric plexus. Whenever necessary, the surgery included adhesiolysis, ovarian surgery, removal of the involved peritoneal tissues, dissection of parametrial planes, isolation of ureteral course (ureterolysis), lateral parametrectomy, posterior parametrectomy, deep uterine vessels identification, colpectomy, bowel resection, etc. The fundamental proposal of nerve-sparing surgery has been the least possible damage to the pelvic innervation in order to reduce postoperative bladder, rectal, and vaginal dysfunction [25].

During laparoscopic exploration of the abdominal cavity (regardless if robot-assisted or not), the lesions previously identified by physical examination and MRI were assessed and resected. To explore for and address complex endometriotic lesions involving the bladder or ureter intraoperative cystoscopy was performed systematically, Hysteroscopy to address intra-uterine conditions and the treatment of other diseases not associated with endometriosis (i.e. cholecystectomy, hernia repair) were performed intraoperatively in some cases, as indicated.

The bladder (Foley) catheter was removed once the residual urine volume was consistently <100 mL. After discharge, patients were seen monthly by the multidisciplinary team (gynecologist, proctologist, urologist, psychologist and nutritionist) until the sixth month, when they were referred back to their regular gynecologist.

## Statistics

The Nonparametric Mann-Whitney U test was used to compare two independent samples (ordinal variables). Chi-square analyses were conducted in the comparison of groups on categorical data (Fisher exact test was used when tables had low expected frequencies). The bivariate Spearman's rank correlation analysis (rho) was used to express the strength of association between two ordinal variables. Tables and statistics were developed using IBM® SPSS® Statistics Standard Grad Pack 20 (IBM Corp., Armonk, NY, USA). The statistical results were considered significant when P<0.05 (2-sided).

## Results

The median follow-up time was 6.3 months (25th percentile: 6.1; 75th percentile: 6.9; minimum: 5.3; maximum: 9.9). No subjects dropped out of the study or declined post-operative reassessment.

Of the 126 women who enrolled in this study, five women reported no vaginal intercourse during the six-month follow-up period (personal reasons not associated with pain) and were thus excluded from the analysis (Fig 1). Considering the symptoms prior to surgery, the dyspareunia severity (primary outcome) showed a statistically significant association with dysmenorrhea and with nonmenstrual/acyclic pelvic pain. The Spearman's rank correlation coefficient (rho) between dyspareunia and dysmenorrhea was 0.240 (P = 0.014; N = 103) and between dyspareunia and nonmenstrual/acyclic pelvic pain was 0.260 (P = 0.004; N = 121).

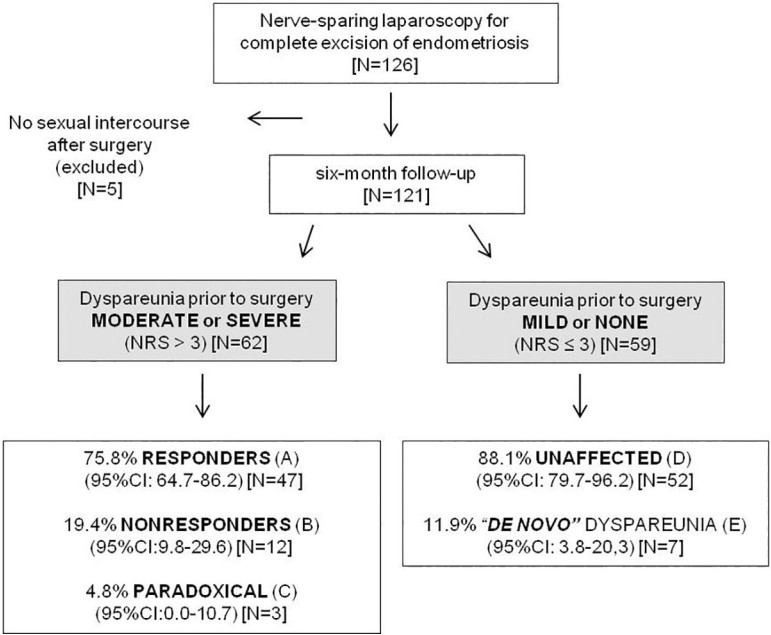

**Fig 1. Response of deep dyspareunia intensity (primary outcome) to nerve-sparing laparoscopy.** Deep dyspareunia was assessed on a self-reported 11-point numeric rating scale (NRS) on two occasions: prior to surgery (during the preoperative evaluation period) and at six-month follow-up. (A) NRS ↓ by 3 points or more. (B) NRS changed by less than 3 points. (C) NRS ↑ by 3 points or more. (D) NRS ≤ 3 both before and after surgery [42 women reported no dyspareunia (NRS = 0) both before and after surgery 71.2% (95%CI: 59.4–82.8)]. (E) NRS prior to surgery = 0 and NRS ↑ by 3 points or more. According to the scores, dyspareunia was hierarchically categorized as none/mild (0–3), moderate/tolerable (4–6), and severe/terrific dyspareunia (7–10).

The number of observations for dysmenorrhea assessment was less than 121, as 17 women taking hormone therapy to suppress menstruation and one woman with previous hysterectomy were excluded from this analysis.

When the total sample (N = 121) was considered, the dyspareunia improvement six months after surgery was statistically significant (P< 0.001). At six-month follow-up, a total of 97 women reported no or mild dyspareunia (NRS< = 3; 80.2%; 95%CI: 72.7–87.6) of which 84 women were absolutely free from dyspareunia (NRS = 0; 69.4%; 95%CI: 61.2–77.7). The median dyspareunia NRS scores (25–75th percentiles) reported prior to surgery and at six-month follow-up were 4 (0–7) and 0 (0–3), respectively.

The total sample was then grouped according to the preoperative dyspareunia scores and the responses of deep dyspareunia intensity (primary outcome) to nerve-sparing laparoscopy were summarized in Fig 1. There were few statistically significant differences with regard to demographic characteristics for women who did not experience dyspareunia or for whom it was mild, versus women whose dyspareunia was graded as moderate or severe. The median age of women with moderate or severe dyspareunia was higher (P = 0.004) and a higher proportion were Caucasian (P = 0.031) (Table 1).

When the 62 women with preoperative moderate/severe dyspareunia (NRS>3) were assessed separately, the dyspareunia decreased by 3 points or more in 47 cases (75.8%; 95%CI: 64.7–86.2) and, among these cases (labeled "**Responders**"), the dyspareunia resolved completely (NRS = 0) in 37 cases (59.7%; 95%CI: 47.8–71.6). The dyspareunia was only somewhat changed (2 points or less; up or down) in 12 cases (19.4%; 95%CI: 9.8–29.6); these subjects were considered "**Nonresponders**". The dyspareunia unexpectedly worsened by 3 points or more in three women (4.8%; 95%CI:0–10.8); their scores increased from 4 to 9, 5 to 8, and 6

**Table 1. Demographic characteristics prior to surgery (N = 121).**

| | | NRS>3 | N = 62 | | NRS< = 3 | N = 59 | | |
|---|---|---|---|---|---|---|---|---|
| | | N | % | | N | % | | P value |
| **Ethnicity** | Caucasian | 34 | 54.8 | | 47 | 79.7 | | 0.031 |
| **(self reported)** | Indian | 1 | 1.6 | | 0 | 0 | | |
| | African | 10 | 16.1 | | 5 | 8.5 | | |
| | Mixed | 17 | 27.4 | | 7 | 11.9 | | |
| **Partner** | Yes | 51 | 82.3 | | 47 | 79.7 | | 0.914 |
| **(stable relationship)** | Not currently | 5 | 8.1 | | 6 | 10.2 | | |
| | Never | 6 | 9.7 | | 6 | 10.2 | | |
| **Schooling** | High school | 14 | 22.6 | | 9 | 15.3 | | 0.538 |
| **(completed degree)** | College | 20 | 32.3 | | 23 | 39.0 | | |
| | Post-grad | 28 | 45.2 | | 27 | 45.8 | | |
| **Income (US$/year)** | Up to 10,000 | 9 | 14.8 | [1] | 8 | 13.6 | | 0.979 |
| | 10 to 25,000 | 22 | 36.1 | | 22 | 37.3 | | |
| | >25,000 | 30 | 50.8 | | 29 | 49.2 | | |
| **Smoking** | Never | 53 | 86.9 | [1] | 55 | 94.8 | [1] | .0119# |
| | Currently or in the past | 8 | 13.1 | | 3 | 5.2 | | |
| **Alcohol consumption** | No | 19 | 32.2 | [2] | 16 | 27.6 | [1] | 0.856 |
| | Up to once a week | 31 | 52.5 | | 33 | 56.9 | | |
| | Twice a week or more | 9 | 15.3 | | 9 | 15.5 | | |
| **Physical activity** | No | 32 | 51.6 | | 31 | 53.4 | [1] | 0.078 |
| | Once or twice a week | 14 | 22.6 | | 5 | 8.6 | | |
| | 3 times a week or more | 16 | 25.8 | | 22 | 37.9 | | |
| | | 25th | Median | 75th | 25th | Median | 75th | |
| **Weight (Kg)** | | 55 | 63 | 74 | 57 | 66 | 75 | 0.237 |
| **Height (cm)** | | 157 | 162 | 167 | 158 | 163 | 167 | 0.674 |
| **BMI (Kg. m$^{-2}$)** | | 20.8 | 23.6 | 27.6 | 22.2 | 24.2 | 26.6 | 0.374 |
| **Age (years)** | | 31.5 | 34.2 | 38.5 | 34.3 | 38.2 | 42.8 | 0.004 |

NRS: self-reported 11 points (0–10) numeric rating scale. Nonparametric independent-samples Mann-Whitney U test was used to compare groups (ordinal variables). Pearson Chi-square test was used to compare categorical variables (#Fisher's Exact test). Number of cases with Missing data between brackets (not answered). Income: Annual Household Income estimated in May 2020. No alcohol consumption includes 4 women who stopped drinking for over 1 year.

The women were grouped according to the severity of the dyspareunia reported prior to surgery: NRS>3 (moderate/severe) or NRS< = 3 (none/mild).

to 10. Their response to treatment is labeled as "**Paradoxical**". The probability of moderate or severe dyspareunia worsening more than 2 points within six months of surgery is estimated as 4.8% (95%CI: 0.0–10.7). The group of women who presented moderate or severe dyspareunia before the surgery showed a significant improvement after surgery (P < .001) (Fig 2A).

Of the 59 women reporting none or mild preoperative dyspareunia (NRS< = 3), 52 still reported mild or no dyspareunia after six months (88.1%; 95%CI: 78.8–96.1); these patients were labeled as "**Unaffected**". Considering only the 52 women who reported no previous dyspareunia (NRS = 0), the scores reported six months after surgery remained 0 in 42 of them (80.8%; 95%CI: 69.6–91.7). Among the 10 women whose scores increased from zero, 3 of them developed mild dyspareunia (NRS = 1, 2 and 3) that was not considered a major problem. However, four women developed moderate dyspareunia (NRS = 4, 4, 5, and 6), and three women developed severe dyspareunia (raw scores 7, 8, and 8). Thus, concerning the six-month follow-up of the 52 women without any previous dyspareunia (NRS = 0), their probability of reporting "**de novo**" moderate and severe dyspareunia could be estimated as 7.7%

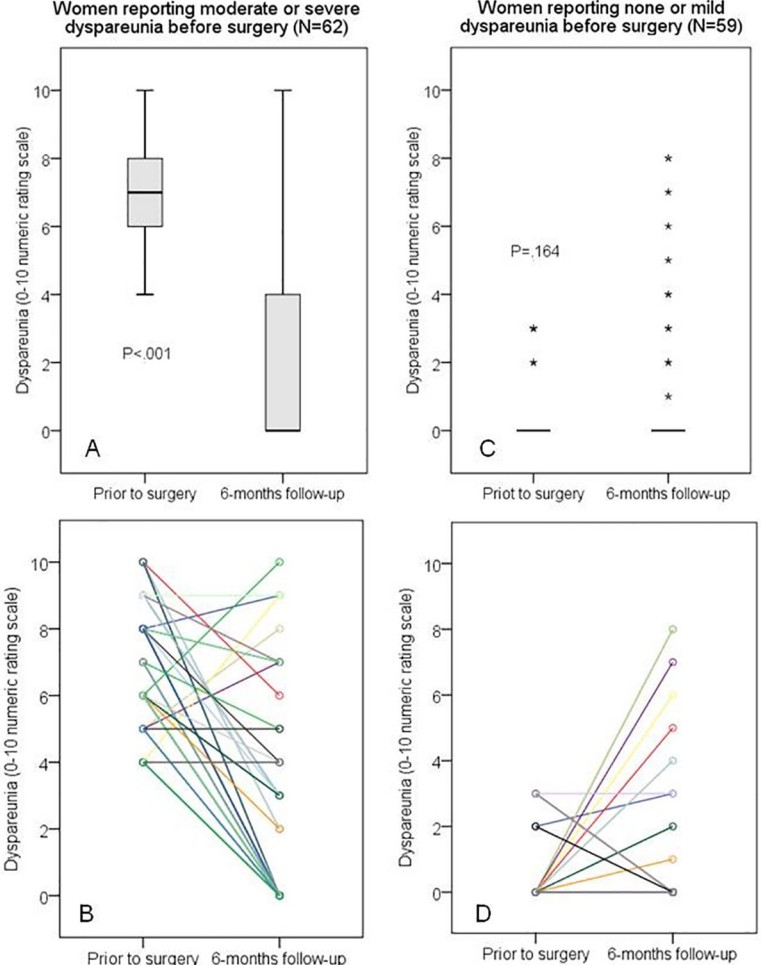

**Fig 2. The severity of deep dyspareunia (primary outcome).** It was assessed on a self-reported 11-point numeric rating scale (NRS) in two moments: prior to surgery (during the preoperative evaluation period) and at six-month follow-up (N = 121). (A) and (C): Boxplots show the 25th, 50th, and 75th percentiles; nonparametric independent-samples Mann-Whitney U test was used for comparison (P < .001); asterisks were used to identify extreme outliers (values higher than 3 x interquartile range). (B) and (D): Point to point lines highlighting the different individual changes; the lines were colored just to emphasize being different women (coincidentally, some cases generated overlapping lines).

(95%CI: 1.8–15.8), and 5.8% (95%CI: 0.0–13.0), respectively. The dyspareunia scores of the women who presented none or mild dyspareunia before the surgery showed no significant change after surgery (P < .164) (Fig 2C).

To identify and then attempt to explain different outcome profiles regarding the response to surgery–based on the dyspareunia score at six-month follow-up in relation to the dyspareunia score preoperatively–the cohort was categorized in subgroups (Tables 2–5). We were especially interested in conducting a more detailed analysis of three of these subgroups: the cases in which the surgery did not achieve the expected dyspareunia relief (Nonresponders), the cases in which dyspareunia paradoxically worsened, and the cases in which dyspareunia arose "de novo" in endometriosis patients whose surgery was indicated for other issues (e.g. infertility). However, because the number of cases in several of these subgroups was too small to attain statistical significance, we recognized that a more qualitative perspective had to be considered and additional individual information had to be analyzed. That individual detail included

**Table 2. Demographic characteristics, personal habits and obstetric history of the 22 women who were not considered responders for deep dyspareunia.**

| Case | Age | Ethnicity | BMI | Partner | Schooling | Regular job | Income | Smoking | Alc | Phys | Menarche | G | P | A | C |
|------|-----|-----------|-----|---------|-----------|-------------|--------|---------|-----|------|----------|---|---|---|---|
| **Nonresponder** | | | | | | | | | | | | | | | |
| 1 | 34.4 | Cauc | 26.6 | Yes | High school | housewife | >25 | 1,5 | 1 | 2 | 12 | 2 | 2 | 0 | 2 |
| 2 | 31.7 | Cauc | 19.7 | Yes | College | economist | >25 | 2 | 1 | 4 | 10 | 0 | 0 | 0 | 0 |
| 3 | 32.8 | Cauc | 30.3 | Yes | High school | hair stylist | 10–25 | 0 | 0 | 2 | 13 | 1 | 0 | 1 | 0 |
| 4 | 38.4 | Cauc | 25.3 | Yes | College | executive | >25 | 0 | 0 | 5 | 15 | 1 | 1 | 0 | 1 |
| 5 | 37.7 | Mix | 22.5 | Yes | Post-grad | nurse | >25 | 0 | 0 | 0 | 12 | 1 | 0 | 1 | 0 |
| 6 | 37.9 | Mix | 28.7 | No | College | executive | <10 | 0 | 1 | 0 | 13 | 1 | 1 | 0 | 1 |
| 7 | 42.9 | Cauc | 21.1 | Yes | High school | flight attendant | >25 | 0 | 1 | 2 | 12 | 0 | 0 | 0 | 0 |
| 8 | 43.9 | Cauc | 24.4 | Yes | Post-grad | lawyer | 10–25 | 0 | 0 | 0 | 13 | 0 | 0 | 0 | 0 |
| 9 | 38.8 | Afric | 30.8 | Yes | High school | salesperson | >25 | 0 | 0 | 0 | 12 | 0 | 0 | 0 | 0 |
| 10 | 32.8 | Cauc | 20.9 | Yes | Post-grad | event promoter | >25 | 0 | 2 | 0 | 12 | 0 | 0 | 0 | 0 |
| 11 | 33.2 | Cauc | 21.5 | Yes | Post-grad | nurse | 10–25 | 0 | 1 | 3 | 13 | 0 | 0 | 0 | 0 |
| 12 | 28.3 | Cauc | 20.6 | Yes | College | actress | >25 | 0 | 1 | 3 | 11 | 0 | 0 | 0 | 0 |
| **Paradoxical** | | | | | | | | | | | | | | | |
| 13 | 46.3 | Mix | 22.6 | Yes | High school | secretary | <10 | 0 | 2 | 0 | 12 | 1 | 1 | 0 | 1 |
| 14 | 51.3 | Cauc | 32.6 | Yes | Post-grad | speech therapist | 10–25 | 4,5 | 1 | 0 | 9 | 2 | 2 | 0 | 2 |
| 15 | 28.4 | Cauc | 20.5 | Yes | Post-grad | nurse | 10–25 | 0 | 1 | 3 | 12 | 0 | 0 | 0 | 0 |
| **"de novo"** | | | | | | | | | | | | | | | |
| 16 | 35,2 | Cauc | 24.5 | Yes | College | businesswoman | >25 | 0 | 0 | 3 | 12 | 1 | 1 | 0 | 1 |
| 17 | 29.4 | Cauc | 22.9 | Yes | College | lawyer | 10–25 | 0 | 0 | 5 | 9 | 0 | 0 | 0 | 0 |
| 18 | 40.4 | Cauc | 23.5 | No | Post-grad | manager | >25 | 0 | 1 | 3 | 14 | 1 | 1 | 0 | 1 |
| 19 | 49.9 | Cauc | 24.2 | Yes | High school | manager | 10–25 | NI | 1 | 0 | 13 | 2 | 2 | 0 | 2 |
| 20 | 44.1 | Cauc | 24.6 | Yes | Post-grad | social worker | <10 | 0 | 1 | 0 | 9 | 4 | 3 | 1 | 3 |
| 21 | 33.7 | Cauc | 29.8 | Yes | College | engineer | >25 | 0 | 1 | 3 | 13 | 0 | 0 | 0 | 0 |
| 22 | 39.8 | Cauc | 20.8 | Yes | Post-grad | lawyer | >25 | 0 | 1 | 3 | 13 | 1 | 1 | 0 | 1 |

Age at surgery (years). BMI: body mass index in Kg.m-2. Partner: stable relationship (self-reported). Ethnicity (self-reported), Cauc: Caucasian, Afric: African ancestral origin, Mix: mixed. Schooling: highest completed degree. Income: annual Household Income estimated in May 2020 (US$ x 1000/year). Smoking: smoking pack years (NI means "not informed"). Alc: alcohol intake frequency (times a week). Phys: physical activity frequency (times a week). Menarche: age in years. G: number of gestations, P: number of births, A: number of abortions, C: number of caesarean-sections. The cases were ordered by severity of dyspareunia reported prior to surgery.

demographic characteristics (Table 2), some possible confounders (Table 3), the history of specific prior surgical procedures (Table 4), and the different classifications of endometriosis (Table 5).

The main findings in the Nonresponders subgroup (12 patients) were: previous pelvic surgery for endometriosis (0); some diagnosed mental disorder (7); no hormone therapy after surgery (6); bilateral uterosacral ligament resection (12); bilateral/unilateral parametrium resection (6/4); colporrhaphy (7); retrocervical (7) and rectovaginal (8) endometriosis nodule excision. For the Paradoxical subgroup (3 patients), the main findings were: previous vaginal hysterectomy (1); some diagnosed mental disorder (2); no hormone therapy or goserelin after surgery (2); bilateral uterosacral ligament resection (2); bilateral/unilateral parametrium resection (1/1); colporrhaphy (3); retrocervical (1) and rectovaginal (2) endometriosis nodule excision. In the subgroup of women reporting "de novo" dyspareunia (7 patients), the main findings included: previous pelvic surgery for endometriosis (3); some diagnosed mental disorder (4); no hormone therapy or goserelin after surgery (5); bilateral/unilateral uterosacral ligament resection (6/1); bilateral/unilateral parametrium resection (2/2); colporrhaphy (5); retrocervical (5) and rectovaginal (4) endometriosis nodule excision.

**Table 3. Possible confounders in assessing pain response to surgery and the changes in self-reported endometriosis-related symptoms assessed prior to surgery and at follow-up in the 22 women who were not considered responders for deep dyspareunia.**

| Case | Dysp (NRS) | Dysm (NRS) | PPain (NRS) | Length of NM PPain | Pelvic surgery in the past | Mental disorder | Indication of surgery | Follow-up (months) | Hormones prior to surgery | Hormones at follow-up | Goserelin implant |
|------|------|------|------|------|------|------|------|------|------|------|------|
| **Nonresponder** | | | | | | | | | | | |
| 1 | 9 → 9 | Ø → Ø | 9 → 2 | 36 | | | P | 6.2 | OP | No | 0 |
| 2 | 9 → 7 | 7 → 7 | 5 → 6 | 24 | | | P | 6.9 | OP+COC | cCOC | 0 |
| 3 | 9 → 7 | 9 → Ø | 8 → 5 | 12 | | mood+sleep | P+I | 7.1 | OP | OP | 0 |
| 4 | 8 → 9 | 0 → Ø | 10 → 8 | 3 | | | P | 5.3 | VR | VR | 0 |
| 5 | 8 → 7 | 8 → Ø | 8 → 4 | 72 | | not assessed | P+I | 6.4 | No | OP | 2 |
| 6 | 8 → 7 | 10 → Ø | 5 → 2 | 30 | | anxiety | P | 6.1 | OP | No | 0 |
| 7 | 7 → 5 | 7 → 0 | 5 → 2 | 24 | | mood+anxiety | P | 6.6 | No | No | 2 |
| 8 | 6 → 4 | 10 → 3 | 4 → 4 | 6 | | sleep | P+I | 6.4 | No | No | 0 |
| 9 | 5 → 7 | 7 → 5 | 8 → 0 | 24 | | | I | 7.0 | No | No | 0 |
| 10 | 5 → 5 | 10 → 8 | 4 → 4 | 24 | | relational | P+I | 6.1 | OP | No | 0 |
| 11 | 4 → 4 | 1 → 3 | 0 → 0 | 0 | | | I | 6.3 | No | No | 0 |
| 12 | 4 → 4 | Ø → Ø | 0 → 0 | 0 | | mood | P | 6.1 | OP+COC +IUD | IUD | 0 |
| **Paradoxical** | | | | | | | | | | | |
| 13 | 6 → 10 | 9 → Ø | 5 → 3 | 6 | VagHTM | anxiety+panic | P | 6.0 | COC | No | 0 |
| 14 | 5 → 8 | 9 → Ø | 8 → 0 | 3 | | mood | P | 6.0 | No | No | 0 |
| 15 | 4 → 9 | 8 → 5 | 5 → 5 | 3 | | | I | 6.0 | No | No | 0 |
| **"de novo"** | | | | | | | | | | | |
| 16 | 0 → 8 | 7 → Ø | 8 → 0 | 20 | | | P | 8.5 | No | No | 1 |
| 17 | 0 → 8 | 10 → 5 | 4 → 0 | 72 | | mood | P+I | 6.1 | No | No | 0 |
| 18 | 0 → 7 | 8 → Ø | 0 → 0 | 0 | Opt+End | | P | 5.9 | COC | cCOC | 0 |
| 19 | 0 → 6 | Ø → Ø | 10 → 3 | 3 | SubHTM+End | pain (fibromyalgia) | P | 6.4 | No | No | 0 |
| 20 | 0 → 5 | 8 → Ø | 0 → 5 | 0 | | | P | 7.1 | No | No | 0 |
| 21 | 0 → 4 | 8 → 2 | 6 → 6 | 2 | | eating+sleep | P+I | 6.9 | No | No | 0 |
| 22 | 0 → 4 | 3 → Ø | 9 → 0 | 144 | Opt+Otm+End +A | anxiety+mood | P+I | 6.3 | OP | cCOC | 0 |

NRS: self-reported 11 points (0–10) numeric rating scale prior to surgery (during the preoperative evaluation period) and at six-months follow-up.

Nonresponders: NRS changed by less than 3 points. Paradoxical response: NRS ↑ by 3 points or more. "de novo" dyspareunia: NRS prior to surgery = 0 and NRS ↑ by 3 points or more. Dysp: deep dyspareunia. Dysm: dysmenorrhea (Ø means not assessed—hormone blockade). PPain: nonmenstrual pelvic pain. NM: nonmenstrual (acyclic). Length of nonmenstrual pelvic pain (in months). Pelvic surgery in the past includes VagHTM: vaginal hysterectomy, Opt: oophoroplasty, Otm: oophorectomy, SubHTM: subtotal hysterectomy, End: endometriosis, A: Arthrodesis (spinal fusion)].

Mental disorders according the International Classification of Diseases published by the World Health Organization; diagnosis was established by a psychologist and/or a psychiatrist. Case 5 was not assessed by a psychologist. The pain disorder of the case 19 includes fibromyalgia. Case 21's eating disorder is binge eating. Indications of surgery inlcudes P: pain, I: infertility issues. Hormone use (those used in the last 3 months) includes COC: combined estrogen-progestin oral contraceptives (cCOC: continuous). VR: combined estrogen-progestin vaginal ring. IUD: progestin intrauterine device. Goserelin: 10.8 mg goserelin acetate implant (number of injections after surgery). Cases 12 and 22 were the only who underwent long-term postoperative physical therapy as a complementary treatment (the positive outcomes subsequent to the six-month follow-up were not included in this study). No patient required bladder catheterization for more than 2 weeks.

## Discussion

Considering the entire cohort (N = 121), the study found statistically significant improvements in dyspareunia at six-month follow-up. The median dyspareunia score improvement was 4 points, and more than two-thirds of the women attained the best possible result, which is to be

**Table 4. Systematic description of the surgical procedures concerning the main sites of endometriosis in the 22 women who were not considered responders for deep dyspareunia.**

| Case | Tube (L/R) | Ovary (L/R) | Uterus | Round Lig | Bladder | Uret | Param (L/R) | Nerve (L/R) | UsLig | Vag | Rcerv | RvSept | Bowel | Barr | Length (min) |
|---|---|---|---|---|---|---|---|---|---|---|---|---|---|---|---|
| **Nonresponder** | | | | | | | | | | | | | | | |
| 1 | S/S | F/PexF | Ht | | | | /X | | LR | H | N.A. | X | | F | 105 |
| 2 | P/B | F/F | MAS | | | R | /Xd | | LR | H | X | | | C | 200 |
| 3 | P/B | CF/F | PLig | L | P | LR | Xd/Xd | Hn/ | LR | X | X | X | Shav | C | 102 |
| 4 | P/P | DF/DF | | | Fb | L | Xd/ | Hn/ | LR | V | X | X | | C | 117 |
| 5 | B/P | F/EPF | M | | | | Xd/ | | LR | | X | X0 | Ap+Seg | F | 185 |
| 6 | S/S | F/F | Ht | LR | Sut | | Xd/X | Hn/ | LR | V | N.A. | X | Seg | F | 281 |
| 7 | BH/P | F/F | MSLig | | | | X/Xd | | LR | H | X | X | | C | 147 |
| 8 | P/P | | MAS | | | | X/X | | LR | | | X | | C | 164 |
| 9 | B/P | /D | MP | | | | | | LR | | | | | | 60 |
| 10 | P/P | F/F | | | P | | X/Xd | | LR | | X | X | Disc | F | 122 |
| 11 | P/P | /F | | | | | | | LR | V | X | | | F | 56 |
| 12 | P/B | | | | | | Xd/Xd | N/Hp | LR | | | X | | C | 99 |
| **Paradoxical** | | | | | | | | | | | | | | | |
| 13 | N.A. | PexF/ | N.A. | LR | | | | | | H | N.A. | X | | C | 75 |
| 14 | S/S | /CF | Ht | LR | | | /X | | LR | H | N.A. | | | C | 112 |
| 15 | P/P | F/F | A | | | LR | Xd/Xd | | LR | H | X | X | Shav | C | 129 |
| **"de novo"** | | | | | | | | | | | | | | | |
| 16 | P/B | Lig/ | MSLig | LR | | | | | LR | | X | | | C | 91 |
| 17 | N/H | EPF/EPF | | R | | | X/ | | LR | H | X | X | Ap | C | 126 |
| 18 | S/N | TF/ | ASLig | LR | | | | | LR | H | X | | | F | 124 |
| 19 | S/S | | N.A. | | | R | /X | | R | H | N.A. | | | C | 85 |
| 20 | S/S | | Ht | | | | | | LR | H | N.A. | X | | F | 83 |
| 21 | P/P | EPF/EPF | A | L | P | L | Xd/X | N/Hn | LR | X | X | X | DDisc | F | 249 |
| 22 | S/S | F/F | MS | LR | | LR | X/X | Hn/ | LR | | X | X | | | 176 |

Nonresponders: NRS changed by less than 3 points. Paradoxical response: NRS ↑ by 3 points or more. "de novo" dyspareunia: NRS prior to surgery = 0 and NRS ↑ by 3 points or more. L/R: Left/Right. X: endometriotic nodule excision. Tube: P pervious or B blocked at chromopertubation; S salpingectomy; H hydrosalpinx; N.A. not applicable (previous salpingectomy). Ovary: T oophorectomy; P oophoroplasty; D drilling; C simple cyst; E endometrioma; Pex oophoropexy; Lig ligation of the ovarian veins; F presence of endometriosis in the peritoneum of the ovarian fossa. Uterus: Ht total hysterectomy, M myomectomy, A adenomyomectomy; Lig permanent uterine arteries ligation; P hysteroscopic polypectomy; S suture; N.A. not applicable (previous hysterectomy). Round Lig: Round ligament. Bladder: Sut partial cystectomy and intracorporeal suturing; P superficial nodule infiltrating bladder peritoneum "shaving"; Fb foreign body on the vesicouterine septum (retained suture material). Uret means ureterolysis (systematic procedure aimed at exposing the ureter in order to free it from external pressure or adhesions or to avoid injury to it during surgery). Param (Parametrium): Xd means deeper resection, below the ureter (paracolpium). Nerve (excision of endometriosis nodule infiltrating nerve): Hn hypogastric nerve; Hp inferior hypogastric plexus. UsLig: Uterosacral ligament. Vag (Vagina): H horizontal colporrhaphy; V vertical colporrhaphy; X endometriotic nodule excision only (no suture required). Rcerv (Retrocervical area): N.A. not applicable because it was excised with the uterus with or without endometriotic nodule (cases 13 and 19 had previous hysterectomy). RvSept (Rectovaginal septum): X0 means dissected, but without endometriosis. Bowel: Ap appendicectomy; Seg segmental; Shav shaving; Disc discoid; DDisc double discoid resection. Barr (Barrier agents for adhesion prevention): F fibrin sealant (human); C carboxymethylcellulose. Length: Duration of pneumoperitoneum.

completely free of dyspareunia (NRS = 0). The positive response to surgery occurred predominantly in those women who reported preoperative moderate/severe dyspareunia (NRS>3). In these subjects, the median dyspareunia scores improved 7 points (Fig 2A). This finding is consistent with prior studies which found that the severity of pain experienced by women with endometriosis could be used to predict their response to surgery [26]. Furthermore, a case-by-case analysis identified different patterns of response to surgery. Although most cases were

**Table 5. Classification of endometriosis and specific comments about the surgery performed in the 22 women who were not considered responders for deep dyspareunia.**

| Case | Phenotype | ENZIAN | rASRM | Additional information |
|---|---|---|---|---|
| Nonresponder | | | | |
| 1 | P | B1 | 2 | |
| 2 | D | B3FA | 2 | Hysteroscopy and cholecystectomy also performed. |
| 3 | P+D | A2B3C1 | 3 | Adenomyosis on MRI. |
| 4 | D | A1B1 | 1 | |
| 5 | E+P+D | B3C1FAFI | 3 | Endometriosis in appendix and cecum. |
| 6 | P+D | A3B3FB3 | 4 | Excision of nodule involving sacral roots. Protective ileostomy. Postoperative vault hematoma. Adenomyosis on MRI. |
| 7 | D | A2B3FA | 3 | |
| 8 | P+D | A1B3FA | 2 | |
| 9 | D | B1 | 1 | Robotic assisted. |
| 10 | P+D | A1B1C2 | 2 | Adenomyosis on MRI. |
| 11 | D | B1 | 1 | Adenomyosis on MRI. |
| 12 | D | A1B3 | 1 | Nodule involving lower right hypogastric plexus, almost reaching sacral roots. |
| Paradoxical | | | | |
| 13 | D | A3 | NA | Previous vaginal hysterectomy. Major adherence involving rectum and left ovary. |
| 14 | D | B1FA | 2 | |
| 15 | D | A3C1FA | 4 | |
| "de novo" | | | | |
| 16 | D | B1 | 1 | Adenomyosis on MRI. |
| 17 | E+P+D | A2B2FI | 4 | Endometriosis apendicular. Fitz-Hugh-Curtis syndrome. |
| 18 | D | B3FA | 3 | Permanent ligation of the uterine arteries and ascending branch of the right uterine artery. |
| 19 | D | B2 | NA | Robotic assisted. Previous hysterectomy (adnexa preserved bilaterally). |
| 20 | P+D | A1B3 | 2 | Postoperative infected vault hematoma. Adenomyosis on MRI. |
| 21 | E+P+D | A3B3C3FAFI | 4 | Robotic assisted. Cholecystectomy also performed. |
| 22 | P+D | A3B3FA | 4 | Robotic assisted. |

Nonresponders: NRS changed by less than 3 points. Paradoxical response: NRS ↑ by 3 points or more. "de novo" dyspareunia: NRS prior to surgery = 0 and NRS ↑ by 3 points or more. Phenotype: P peritoneal endometriosis; E endometrioma; D lesions deeper than 5 mm. ENZIAN: Revised classification (2012). rASRM: Revised American Society of Reproductive Medicine staging classification (1996). Adenomyosis on MRI: Adenomyosis was reported by a radiologist who reviewed the Magnetic Resonance Imaging, but was not surgically explored and thus not included in ENZIAN classification.

classified as Responders, some women showed no improvement of their dyspareunia and, in fact, there was a small subgroup in which dyspareunia actually worsened (dubbed "Paradoxical" dyspareunia) or started to occur ("de novo" dyspareunia).

Although statistically significant, the correlations between dyspareunia and dysmenorrhea and between dyspareunia and nonmenstrual/acyclic pelvic pain were weak prior to surgery. These findings are consistent with previous studies and support both the heterogeneous nature of endometriosis and the existence of covariates affecting women's perception of different types of pain [1, 10].

## Anatomical considerations

The uterosacral ligaments are thought to be the most common anatomical site of deep infiltrating endometriosis [27] and they are associated with deep dyspareunia [28]. Dyspareunia may be induced by tension on the affected uterosacral ligaments during intercourse [29] and the intensity of symptoms seems to correlate with the extent of endometriotic lesions infiltrating

the ligaments [29, 30]. Most cases of dyspareunia have endometriotic lesions infiltrating the uterosacral ligaments [31], but few studies have investigated the consequences of the surgical treatment for endometriosis in the uterosacral ligaments [32]. Although asymptomatic women (NRS = 0) were not included in their study, Montanari et al. [21] found statistically significant associations between involvement of ENZIAN compartment B (uterosacral ligaments, parametrium) and presence of dyspareunia, and between dyspareunia scores and lesion size in ENZIAN compartment B. In the present study, of the 22 cases considered to have poor outcomes for dyspareunia, 20 had undergone bilateral complete resection of endometrial lesions in the uterosacral ligaments and the resection had been unilateral in one. In other words, only one of the 22 women with poor outcome did not have endometriosis in both uterosacral ligaments. The existence of endometriosis in the uterosacral ligaments could explain some degree of preoperative dyspareunia, but could not explain dyspareunia after surgery. The same reasoning applies to the parametrium, which was resected in 16 of these cases, and bilaterally in seven cases. Although excision and ablation of endometrial lesions in the uterosacral ligaments may have distinct surgical results for dyspareunia in women with minimal to mild endometriosis [33], no patient in this series underwent ablation only. Besides, lateral parametrial endometriosis is a condition that reflects a more severe manifestation of endometriosis and usually requires more aggressive surgery [34], which may have a greater impact on nervous dysfunctions and, consequently, on sexual function.

The presence of vaginal endometriotic lesions also has been associated with severe dyspareunia [35] and rectovaginal nodules have been found to be associated with more impaired sexual activity and more sexual dysfunction [20]. However, few studies have investigated complication rates associated with treatment for endometriosis in the rectovaginal septum [32]. In a minority of women, pain during intercourse is one of the more long-lasting sequelae of the hysterectomy [36]. The vaginal vault traumatic neuromas–disorganized proliferation of nerves due to the cell's inability to self-repair in response to injury–are a documented albeit seldom reported cause of vaginal pain following hysterectomy [37] and colporrhaphy [38]. In the present study, of the 22 cases considered poor outcomes, about two-thirds (14) had previously undergone some type of colporrhaphy, which thus could be considered an independent risk factor.

Symptoms associated with adenomyosis include dyspareunia [39]. Adenomyosis may have contributed to dyspareunia in some of the 22 cases with poor dyspareunia response because 15 were diagnosed with adenomyosis preoperatively (MRI) and/or postoperatively (histopathology) (Table 5). Eleven of these 15 women had their uterus preserved (Table 4).

Recurrent dyspareunia [40] and persistent dyspareunia [41] may also occur after excision of retrocervical endometriosis. Postoperative adhesion formation is thought to be the leading cause of such pain [36]. In this series, more than half of the cases with poor dyspareunia response (13 out of the 22) underwent retrocervical nodule excision and major adhesions may have occurred in many of these.

About half of the patients who had an unfavorable dyspareunia outcomes had mild grades of endometriosis, specifically rASRM classification 1 and 2 (Table 5). This observation highlights the importance of careful reflection about whether to indicate surgery over other treatment options for less severe endometriosis, especially when dyspareunia is the primary symptom central to the discussion.

## Issues beyond the endometriotic lesions

There is an appreciation that pain mechanisms in endometriosis extend beyond the presence of endometriotic lesions alone [42]. More precise individualized treatment for endometriosis

is warranted in the setting of overlapping pain and mental health conditions [43]; these will require multidisciplinary expertise [44]. In this series, for example, more than half (13 of 22) of the cases with poor dyspareunia response had a previously diagnosed mental health disorder, which may make some of these patients' subjective assessments more complex.

Women affected by deep infiltrating endometriosis, compared to women with isolated ovarian endometriosis, presented more severe pelvic floor muscle dysfunctions [45]. Even when the surgery is appropriately chosen, well-timed, and the minimally invasive technique performed meticulously, changes in sexual response or in muscle tone may occur, which require further treatment such pelvic floor physical therapy (PT) for residual muscle pain [46]. Indeed, pelvic floor PT has been found to improve pelvic floor muscle relaxation and reduce dyspareunia in women with endometriosis-related pain [45]. In this study, postoperative pelvic floor PT was indicated on a case-by-case basis, and was not evaluated in a clinical trial. Only two cases (12 and 22) with postoperative dyspareunia underwent long-term postoperative pelvic floor PT (Table 3). The late positive outcomes of these two cases were not included in this analysis.

Although the bowel and bladder complications appear to be acceptable and often reversible, surgeons and women have to be aware of the incidence postoperative sexual dysfunctions, such as anorgasmia and insufficient vaginal lubrication, which may persist over time, for example, after segmental resection for bowel deep infiltrating endometriosis [47]. Actually, insufficient vaginal lubrication following surgery for deep infiltrating endometriosis is reported in about one-third, which is a significantly higher percentage than reported by healthy women [32]. Although insufficient vaginal lubrication–part of the excitement/arousal phase of the sexual response cycle–and the late expansion of the upper two-thirds of the vagina may be associated with dyspareunia [33], they were not considered in this study.

Holt et al. [48] report that several important differences in factors related to sexual satisfaction also occur as a function of sexual identity. Although the present study assessed dyspareunia in participants whose sexual function could encompass both opposite-sex and same-sex sexual interactions, a complete sexual function assessment can and should include much more. There are several instruments available to assess female sexual function, which contemplate not only the presence or absence of pain, but also other specific domains, such as desire, arousal, lubrication, orgasm, frequency of sexual activity, pleasure, satisfaction, as well as intimacy and relationship commitment. Specific instruments currently in use include the Female Sexual Function Index [49, 50], the sexual relationship supplementary module of the Endometriosis Health Profile Questionnaire [51], the Personal Well-Being Index [52], and the New Sexual Satisfaction Scale [53], among others.

Last but not least, in the 22 cases in which surgery alone was not sufficient to improve dyspareunia, 13 patients elected to defer any type of post-operative hormonal treatment, most because they wanted to get pregnant (Table 3). Although combined oral contraceptive therapy can have a role in restraining the progression of dysmenorrhea and dyspareunia [54], the choice between clinical (hormonal) or surgical treatment is complex and involves consideration of multiple factors, such as age, reproductive goals, previous pelvic surgery, endometriosis-related impaired quality of life, comorbidities, and each woman's individual concerns about surgery. Both therapeutic approaches can lead to satisfactory diminution of pain. Increasingly, it seems that combined treatment–endometriosis surgery associated with postoperative hormonal blockade–is the therapeutic approach that best minimizes recurrence of pain symptoms and improves quality of life [55, 56]. Although some authors have demonstrated that sexual desire, satisfaction with sex and pelvic problem interference with intercourse may be significantly improved after 6 months from laparoscopic excision of deep endometriosis combined with postoperative combined oral contraceptive therapy [57], postoperative hormonal blockage is not always an option, as least immediately.

## Limitations

Our study has several limitations. The six-month follow-up period may be considered too short there is a need for high-quality prospective studies assessing long-term outcomes [58]. Two conditions associated with pelvic pain were not considered: bladder pain syndrome / interstitial cystitis–the so-called "evil twins" [59]–and irritable bowel syndrome [60]. The possibility of selection bias associated with access to care should be considered. A simple biometric approach for measuring dyspareunia associated with endometriosis is insufficient, and a more elaborate assessment of the effect of the dyspareunia on a woman's sexual function is recommended [20]. Some important variables were not considered in this study, such as sexual satisfaction, desire, arousal, lubrication, orgasm, frequency of sexual activity, pleasure, satisfaction, as well as intimacy and relationship commitment.

Although the changes in dyspareunia in this cohort of 126 patients were reported as insufficient or inadequate in 22 cases, it is clear that most of these women still reported significant improvement in dysmenorrhea and pelvic pain after the surgery (Table 3). Therefore, it is possible that the scores chosen to represent postoperative dyspareunia in some cases have been influenced by changes in these other symptoms.

This was a retrospective analysis of intervention. Nevertheless, the method was stronger than usual retrospective studies because it included a very consistent preplanned process for data capture and the use of validated instruments.

## Statistical power limitations to test hypothesis

The number of cases in each of the three subgroups was too small for the application of specific statistical tests—Nonresponders (12), Paradoxical (3), and "de novo" (7). Therefore, in order to raise hypotheses in relation to possible covariates, we used a balanced approach considering not only the frequencies of the observations, but also the biological plausibility and the information obtained from the literature.

From a qualitative point of view, some factors were then hypothesized to be positive confounders for assessing postoperative dyspareunia. They included: adenomyosis, previous diagnosed mental disorders, lack of hormone therapy after surgery, prior colporrhaphy, and endometriosis nodule excision in the ENZIAN B compartment (uterosacral ligament/parametrium), in the rectovaginal septum or in the retrocervical area. All but one of the 22 cases with poor outcome regarding dyspareunia had at least four of these factors (case 9 had only two). These factors should be considered in future studies and in the counseling of patients about expectations for postoperative outcomes.

Finally, despite being quite common in before-and-after studies, our team recommends being very cautious when comparing median (or mean) scores because endometriosis is a highly-individualized condition, as are results and adverse consequences of its treatment [1]. Indeed, the present study emphasized this phenomenon by focusing on the cases with unexpected (paradoxical) or disappointing outcomes. Overall, endometriosis surgery provides significant improvement in deep dyspareunia. However, patients should be alerted about the possibility of unsatisfactory results.

## Supporting information

**S1 Table. Raw data of dyspareunia assessment.**
(XLSX)

## Acknowledgments

The authors thank Mr. Luiz Fernando Salzano for his advice on information technology and Dr. Leigh J. Passman for reviewing the English manuscript.

## Author Contributions

**Conceptualization:** Claudio Peixoto Crispi Jr., Bruna Rafaela Santos de Oliveira, Marlon de Freitas Fonseca.

**Data curation:** Claudio Peixoto Crispi Jr., Claudio Peixoto Crispi, Bruna Rafaela Santos de Oliveira, Nilton de Nadai Filho.

**Formal analysis:** Claudio Peixoto Crispi Jr., Marlon de Freitas Fonseca.

**Investigation:** Claudio Peixoto Crispi Jr., Bruna Rafaela Santos de Oliveira, Nilton de Nadai Filho.

**Methodology:** Claudio Peixoto Crispi Jr., Claudio Peixoto Crispi, Marlon de Freitas Fonseca.

**Project administration:** Marlon de Freitas Fonseca.

**Software:** Marlon de Freitas Fonseca.

**Supervision:** Claudio Peixoto Crispi, Marlon de Freitas Fonseca.

**Validation:** Fernando Maia Peixoto-Filho, Marlon de Freitas Fonseca.

**Visualization:** Claudio Peixoto Crispi, Fernando Maia Peixoto-Filho, Marlon de Freitas Fonseca.

**Writing – original draft:** Claudio Peixoto Crispi Jr., Bruna Rafaela Santos de Oliveira, Nilton de Nadai Filho, Marlon de Freitas Fonseca.

**Writing – review & editing:** Claudio Peixoto Crispi, Fernando Maia Peixoto-Filho.

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
