## [Decision Letter · Decision Letter 0]

9 Mar 2021

PONE-D-21-03413

SIX-MONTH FOLLOW-UP OF MINIMALLY INVASIVE NERVE-SPARING COMPLETE EXCISION OF ENDOMETRIOSIS: WHAT ABOUT DYSPAREUNIA?

PLOS ONE

Dear Dr. Crispi Jr,

Thank you for submitting your manuscript to PLOS ONE. After careful consideration, we feel that it has merit but does not fully meet PLOS ONE’s publication criteria as it currently stands. Therefore, we invite you to submit a revised version of the manuscript that addresses the points raised during the review process.

We look forward to receiving your revised manuscript.

Kind regards,

Diego Raimondo

Academic Editor

PLOS ONE

Journal Requirements:

3. Please provide additional details regarding participant consent. In the ethics statement in the Methods and online submission information, please ensure that you have specified whether consent was informed.

Reviewers' comments:

Reviewer's Responses to Questions

**Comments to the Author**

1. Is the manuscript technically sound, and do the data support the conclusions?

Reviewer #1: Yes

Reviewer #2: Yes

2. Has the statistical analysis been performed appropriately and rigorously? 

Reviewer #1: Yes

Reviewer #2: Yes

3. Have the authors made all data underlying the findings in their manuscript fully available?

Reviewer #1: Yes

Reviewer #2: Yes

4. Is the manuscript presented in an intelligible fashion and written in standard English?

Reviewer #1: Yes

Reviewer #2: Yes

5. Review Comments to the Author

Reviewer #1: I read with interest this paper and I think is suitable for publication after some major revisions in form and content.

My suggestions are:

- Page 3 and 6, line 61 and 152, please change the phrase “Cytoreductive surgery” in “ eradicating surgery”. Cytoreduction is a term correct for gynaecologic oncology procedures and not for benign conditions.

-Page 4 line 91. It is not necessary anymore to classify with the Canadian task force classification the trials. Please remove it from the text.

-Page 6 line 160. In material and methods no results should be presented in the text. Please remove the sentence “ no patients required bladder …..” and put it in results section.

- I suggest to enlarge the discussion about sexual dysfunction and sexual quality of life after surgery for endometriosis and to report the possible way of treating and diagnosing them. Infact I think in this form there is too much limitation to the only deep dyspaurenia field.

I suggest to find in literature some paper about these issues and among them to read and cite the following:

- Del Forno S, Arena A, Alessandrini M, Pellizzone V, Lenzi J, Raimondo D, Casadio P, Youssef A, Paradisi R, Seracchioli R. Transperineal Ultrasound Visual Feedback Assisted Pelvic Floor Muscle Physiotherapy in Women With Deep Infiltrating Endometriosis and Dyspareunia: A Pilot Study. J Sex Marital Ther. 2020;46(7):603-611. doi: 10.1080/0092623X.2020.1765057. Epub 2020 Jun 24. PMID: 32579077.

-Turco LC, Tortorella L, Tuscano A, Palumbo MA, Fagotti A, Uccella S, Fanfani F, Ferrandina G, Nicolotti N, Vargiu V, Lodoli C, Scaldaferri F, Scambia G, Cosentino F. Surgery-related complications and long-term functional morbidity after segmental colo-rectal resection for deep infiltrating endometriosis (ENDO-RESECT morb). Arch Gynecol Obstet. 2020 Oct;302(4):983-993. doi: 10.1007/s00404-020-05694-0. Epub 2020 Jul 16. PMID: 32676859.

-Mabrouk M, Montanari G, Di Donato N, Del Forno S, Frascà C, Geraci E, Ferrini G, Vicenzi C, Raimondo D, Villa G, Zukerman Z, Alvisi S, Seracchioli R. What is the impact on sexual function of laparoscopic treatment and subsequent combined oral contraceptive therapy in women with deep infiltrating endometriosis? J Sex Med. 2012 Mar;9(3):770-8. doi: 10.1111/j.1743-6109.2011.02593.x. Epub 2012 Feb 9. PMID: 22321207.

Reviewer #2: The paper primary analysed the individual responses of deep dyspareunia six month after nerve-sparing laparoscopic surgery in patients suffering from endometriosis. Secondly the authors analysed dysmenorrhea and acyclic pelvic pain. Very interesting is the qualitative analysis of the several conditions that can impact in cases with poor prognosis.

The article is well written and intelligible.

- Material and methods:

Line 149 the authors point four steps for the diagnosis of endometriosis, but they describe only three.

What does the authors nerve-sparing laparoscopic procedure consists of? which technique do they follow? Any references?

- Results

Although the results are described in tables, the key findings also need to be better described in the text. Please, better clarify line 181-189 and 232-239.

This sentence is repeated many times, it could be summarized “The severity of deep dyspareunia (primary outcome) was assessed on a self-reported 233 11-point numeric rating scale (NRS) in two moments: prior to surgery (during the preoperative evaluation period) and at six-month follow-up (N=121)”.

Regarding the findings in nonresponders/paradoxical/de novo groups I think there is an error in line 334 (bilateral/unilateral parametrium resection: 5/5 instead of 6/4) and in line 341 (bilateral/unilateral uterosacral ligament resection: 7/1 instead of 6/1).

Why did the authors not describe eventual changes of dysmenorrhea and acyclic pelvic pain since they also assessed these symptoms during the study? Did they investigate these symptoms only to find eventual correlation with dysmenorrhea?

- Discussion:

In a previous study (DOI: 10.1016/j.jmig.2018.08.022) lateral parametrial endometriosis has been associated with more aggressive and widespread disease requiring more aggressive surgery, higher rate of associated localizations and seems to have a greater impact on nervous dysfunctions; please elaborate this topic.

Please, the Authors should investigate and clarify how medical therapy might affect dyspareunia. In particular, in a previous study was found that combined oral contraceptive (COC) therapy in women with deep infiltrating endometriosis (before surgery) can restrain the progression of dyspareunia compared to women who did not assume COC. In the latter was shown a worsening of dyspareunia (DOI: 10.1016/j.jmig.2011.04.008). Furthermore two other studies (DOI: 10.1111/j.1743-6109.2011.02593.x; DOI: 10.1136/jfprhc-2014-100993) have shown that the combination of surgical excision of endometriosis and the subsequent COC therapy improves sexual function and all symptoms related to endometriosis (also dyspareunia); moreover the sexual function of women suffering from endometriosis was comparable to that of healthy women. It might be worth examining this aspect.

When the Authors explain that changes in muscle tone may occur, it may be useful to consider that it has been previously demonstrated (DOI: 10.1002/uog.18924) that women affected by deep endometriosis seems to have increased pelvic floor muscle tone than women with isolated ovarian endometriosis; besides, hypertonic dysfunction of pelvic floor can cause pain symptoms and pelvic organ dysfunction potentially resistant to hormonal or surgical therapy for endometriosis.

6. PLOS authors have the option to publish the peer review history of their article (what does this mean?). If published, this will include your full peer review and any attached files.

Reviewer #1: No

Reviewer #2: No

---

## [Author Response · Author response to Decision Letter 0]

26 Mar 2021

Additional requirements:

1. Please ensure that your manuscript meets PLOS ONE's style requirements, including those for file naming. The PLOS ONE style templates can be found at …

We carried out the the Plos Ones style requirements and we believe it is appropriated.

As noted in the “Acknowledgements”, the revised manuscript has been reviewed by a native English speaker (Dr. Leigh J. Passman) who is a physician (a general internist) who himself has published in JAMA, JGIM, Annals of Internal Medicine, Health Affairs, among other journals. He reviewed the entire manuscript including the revised version. He is available to quickly address any remaining concerns raised by the Reviewers or the Editor.

3. Please provide additional details regarding participant consent. In the ethics statement in the Methods and online submission information, please ensure that you have specified whether consent was informed.

We rewrote this part and believe that we address what was requested, such as: “Prospective written consent for inclusion in observational studies was informed and signed by all patients prior to the surgical procedure. These documents are stored at our institute.” (highlighted)

We included this caption at the end (highlighted).

5. Review Comments to the Author 

Reviewer #1: I read with interest this paper and I think is suitable for publication after some major revisions in form and content.

We thank Reviewer #1 for acknowledging our manuscript is “eligible for publication” once his suggestions are were addressed. We believe we have addressed them.

We appreciate Reviewer #1 desire to help us improve the manuscript. Thanks!

 My suggestions are: - Page 3 and 6, line 61 and 152, please change the phrase “Cytoreductive surgery” in “eradicating surgery”. Cytoreduction is a term correct for gynaecologic oncology procedures and not for benign conditions.

The term “cytoreductive” was removed from the manuscript.

-Page 4 line 91. It is not necessary anymore to classify with the Canadian task force classification the trials. Please remove it from the text.

The classification was removed from the manuscript.  -Page 6 line 160. In material and methods no results should be presented in the text. Please remove the sentence “no patients required bladder...” and put it in results section.

This sentence was removed from MAT&MET and placed in RESULTS section (Table 3).  - I suggest to enlarge the discussion about sexual dysfunction and sexual quality of life after surgery for endometriosis and to report the possible way of treating and diagnosing them. Infact I think in this form there is too much limitation to the only deep dyspaurenia field. I suggest to find in literature some paper about these issues and among them to read and cite the following:  - Del Forno S, Arena A, Alessandrini M, Pellizzone V, Lenzi J, Raimondo D, Casadio P, Youssef A, Paradisi R, Seracchioli R. Transperineal Ultrasound Visual Feedback Assisted Pelvic Floor Muscle Physiotherapy in Women With Deep Infiltrating Endometriosis and Dyspareunia: A Pilot Study. J Sex Marital Ther. 2020;46(7):603-611. doi: 10.1080/0092623X.2020.1765057. Epub 2020 Jun 24. PMID: 32579077.   -Turco LC, Tortorella L, Tuscano A, Palumbo MA, Fagotti A, Uccella S, Fanfani F, Ferrandina G, Nicolotti N, Vargiu V, Lodoli C, Scaldaferri F, Scambia G, Cosentino F. Surgery-related complications and long-term functional morbidity after segmental colo-rectal resection for deep infiltrating endometriosis (ENDO-RESECT morb). Arch Gynecol Obstet. 2020 Oct;302(4):983-993. doi: 10.1007/s00404-020-05694-0. Epub 2020 Jul 16. PMID: 32676859.  -Mabrouk M, Montanari G, Di Donato N, Del Forno S, Frascà C, Geraci E, Ferrini G, Vicenzi C, Raimondo D, Villa G, Zukerman Z, Alvisi S, Seracchioli R. What is the impact on sexual function of laparoscopic treatment and subsequent combined oral contraceptive therapy in women with deep infiltrating endometriosis? J Sex Med. 2012 Mar;9(3):770-8. doi: 10.1111/j.1743-6109.2011.02593.x. Epub 2012 Feb 9. PMID: 22321207.

The focus of this paper was not to explore the “sexual function” theme in depth (a much broader and much more complex topic), but only on deep dyspareunia (a relatively common complaint among endometriosis patients). Although extremely pertinent, enlarging the discussion about sexual dysfunction and sexual quality of life in details would make the manuscript very large and, actually, we intend to do this in another paper.

Thus, we

(1) considered these important issues;

(2) enlarged the discussion in the topic “Issues beyond the endometriotic lesions” - (certainly in a superficial way); and

(3) included the suggested papers.

[Obviously, these changes have improved the manuscript. Thanks!]

Regarding the three suggested papers:

1. The publication “Del Forno et al., 2020” had already been considered and cited in the first version of the submitted manuscript (line 421-422).

2. The publication “Turco et al., 2020” was considered and included in the manuscript.

3. The publication “Mabrouk et al., 2012” was considered and included in the manuscript.

Yet, in order to better explore these issues, we also included the references below:

Alkatout I, Mettler L, Beteta C, et al. Combined surgical and hormone therapy for endometriosis is the most effective treatment: prospective, randomized, controlled trial. J Minim Invasive Gynecol. 2013;20(4):473-481.

Holt LL, Chung YB, Janssen E, Peterson ZD. Female Sexual Satisfaction and Sexual Identity. J Sex Res. 2020;1-11.

Pacagnella R de C, Vieira EM, Rodrigues OM Jr, Souza C. Adaptação transcultural do Female Sexual Function Index [Cross-cultural adaptation of the Female Sexual Function Index]. Cad Saude Publica. 2008;24(2):416-426.

Rosen R, Brown C, Heiman J, et al. The Female Sexual Function Index (FSFI): a multidimensional self-report instrument for the assessment of female sexual function. J Sex Marital Ther. 2000; 26(2):191-208.

Jones G, Kennedy S, Barnard A, Wong J, Jenkinson C. Development of an endometriosis quality-of-life instrument: The Endometriosis Health Profile-30. Obstet Gynecol. 2001;98(2):258-264.

Oyanedel JC, Barrientos J, Mella C, et al. Validation of the Sexual Personal Well-being Index (PWI-Sex) [published online ahead of print, 2020 Jun 13]. J Sex Marital Ther. 2020;1-11.

Stulhofer A, Busko V, Brouillard P. Development and bicultural validation of the new sexual satisfaction scale. J Sex Res. 2010; 47(4):257-268.

Thank you for the opportunity to expand (even if only briefly) on this important issue!

Reviewer #2: The paper primary analysed the individual responses of deep dyspareunia six month after nerve-sparing laparoscopic surgery in patients suffering from endometriosis. Secondly the authors analysed dysmenorrhea and acyclic pelvic pain. Very interesting is the qualitative analysis of the several conditions that can impact in cases with poor prognosis. The article is well written and intelligible.

We thank Reviewer #2 for acknowledging our manuscript is “eligible for publication” once his suggestions are were addressed. We believe we have addressed them.

We appreciate Reviewer #2 desire to help us improve the manuscript.

 - Material and methods: Line 149 the authors point four steps for the diagnosis of endometriosis, but they describe only three.

We fixed it.

Thank you very much for identifying this gross error.

What does the authors nerve-sparing laparoscopic procedure consist of? which technique do they follow? Any references? 

Actually, there was a gap; this was really missing.

We added a brief introduction to the concept of nerve-sparing surgery we have followed.

The publication “Ceccaroni et al., 2020” [doi: 10.1016/j.jmig.2019.09.002] was included in the second paragraph of the (renamed) topic “The nerve-sparing surgery” (MAT & METHODS Section). 

Thanks for the suggestion!

 - Results Although the results are described in tables, the key findings also need to be better described in the text. Please, better clarify line 181-189 and 232-239.

We have tried to use the same simple rule that is "You write a figure/table legend so that the reader can fully understand its content without having to refer to the main text."

The lines 181-189 and 232-239 (first version of the submitted manuscript) are the legends of the Figures 1 and 2. They should be used only to a best understanding of the respective Figures.

Actually, we ended up leaving the presentation of the results "too lean" because we were concerned with saving space (minimizing the number of words).

We made some changes to the RESULTS section in order to better present the Figures 1 and 2, which containing some key findings.

We are very grateful for this suggestion!

 This sentence is repeated many times, it could be summarized “The severity of deep dyspareunia (primary outcome) was assessed on a self-reported 233 11-point numeric rating scale (NRS) in two moments: prior to surgery (during the preoperative evaluation period) and at six-month follow-up (N=121)”.

Yes. It is repeated many times. However, we think this is important so that the reader can fully understand the content of every figure and table without having to refer to the main text.

Nonetheless, if this is an absolute requirement, we can obviously make this modification.  Regarding the findings in nonresponders/paradoxical/de novo groups I think there is an error in line 334 (bilateral/unilateral parametrium resection: 5/5 instead of 6/4) and in line 341 (bilateral/unilateral uterosacral ligament resection: 7/1 instead of 6/1).

We fixed these two errors.

Thank you very much for this exceptional review!  Why did the authors not describe eventual changes of dysmenorrhea and acyclic pelvic pain since they also assessed these symptoms during the study? Did they investigate these symptoms only to find eventual correlation with dysmenorrhea?

In this manuscript, the biggest - perhaps the only - reason for us not having explored the changes in dysmenorrhea and acyclic pelvic pain (also prevalent symptoms in endometriosis) was the need to stay focused on a specific subject - a difficult task when dealing with endometriosis.

Soon, we will be evaluating changes in other variables, such as dyschezia, dysuria, dysmenorrhea, acyclic pelvic pain and constipation.

Our work with that cohort is ongoing.

Our learning in the preparation of this original manuscript has been the initial step.  - Discussion: In a previous study (DOI: 10.1016/j.jmig.2018.08.022) lateral parametrial endometriosis has been associated with more aggressive and widespread disease requiring more aggressive surgery, higher rate of associated localizations and seems to have a greater impact on nervous dysfunctions; please elaborate this topic.

The publication “Mabrouk et al., 2019” [doi: 10.1016/j.jmig.2018.08.022] was considered and included in the manuscript in the first paragraph of the DISCUSSION Section – see topic “Anatomical considerations”.

We are very grateful for this suggestion!

 Please, the Authors should investigate and clarify how medical therapy might affect dyspareunia. In particular, in a previous study was found that combined oral contraceptive (COC) therapy in women with deep infiltrating endometriosis (before surgery) can restrain the progression of dyspareunia compared to women who did not assume COC. In the latter was shown a worsening of dyspareunia (DOI: 10.1016/j.jmig.2011.04.008). Furthermore two other studies (DOI: 10.1111/j.1743-6109.2011.02593.x; DOI: 10.1136/jfprhc-2014-100993) have shown that the combination of surgical excision of endometriosis and the subsequent COC therapy improves sexual function and all symptoms related to endometriosis (also dyspareunia); moreover the sexual function of women suffering from endometriosis was comparable to that of healthy women. It might be worth examining this aspect.

We included a brief discussion about hormone therapy, endometriosis and pain. Also, the publications “Mabrouk et al., 2011” [doi: 10.1016/j.jmig.2011.04.008], “Mabrouk et al., 2012” [doi: 10.1111/j.1743-6109.2011.02593.x], and “Di Donato el al., 2015” [doi: 10.1136/jfprhc-2014-100993] were considered and included in the DISCUSSION section [last paragraph of the topic “Issues beyond the endometriotic lesions”].

Thank you for the opportunity to expand (even if only briefly) on this important issue!

 When the Authors explain that changes in muscle tone may occur, it may be useful to consider that it has been previously demonstrated (DOI: 10.1002/uog.18924) that women affected by deep endometriosis seems to have increased pelvic floor muscle tone than women with isolated ovarian endometriosis; besides, hypertonic dysfunction of pelvic floor can cause pain symptoms and pelvic organ dysfunction potentially resistant to hormonal or surgical therapy for endometriosis.

The publication “Mabrouk et al., 2018” [doi: 10.1002/uog.18924] was considered and included in the DISCUSSION section [second paragraph of the topic “Issues beyond the endometriotic lesions”].

Again, thank you for the opportunity to expand (even if only briefly) on this important issue!

---

## [Editor Report · Decision Letter 1]

30 Mar 2021

SIX-MONTH FOLLOW-UP OF MINIMALLY INVASIVE NERVE-SPARING COMPLETE EXCISION OF ENDOMETRIOSIS: WHAT ABOUT DYSPAREUNIA?

PONE-D-21-03413R1

Dear Dr. Crispi Jr,

We’re pleased to inform you that your manuscript has been judged scientifically suitable for publication and will be formally accepted for publication once it meets all outstanding technical requirements.

Kind regards,

Diego Raimondo

Academic Editor

PLOS ONE

Additional Editor Comments (optional):

The authors correctly replied to all reviewers’ queries

Reviewers' comments:

None

---

## [Editor Report · Acceptance letter]

6 Apr 2021

PONE-D-21-03413R1 

Six-month follow-up of minimally invasive nerve-sparing complete excision of endometriosis: what about dyspareunia? 

Dear Dr. Crispi Jr:

I'm pleased to inform you that your manuscript has been deemed suitable for publication in PLOS ONE. Congratulations! Your manuscript is now with our production department. 

Kind regards, 

on behalf of

Dr. Diego Raimondo 

Academic Editor

PLOS ONE